# The Effects of a High-Carbohydrate versus a High-Fat Shake on Biomarkers of Metabolism and Glycemic Control When Used to Interrupt a 38-h Fast: A Randomized Crossover Study

**DOI:** 10.3390/nu16010164

**Published:** 2024-01-04

**Authors:** Landon S. Deru, Elizabeth Z. Gipson, Katelynn E. Hales, Benjamin T. Bikman, Lance E. Davidson, Benjamin D. Horne, James D. LeCheminant, Larry A. Tucker, Bruce W. Bailey

**Affiliations:** 1Department of Exercise Science, Brigham Young University, Provo, UT 84602, USA; 2Division of Physical Activity and Weight Management, University of Kansas Medical Center, Kansas City, KS 66160, USA; 3Department of Cellular Biology and Physiology, Brigham Young University, Provo, UT 84602, USA; 4Intermountain Heart Institute, Intermountain Medical Center, Salt Lake City, UT 84107, USA; benjamin.horne@imail.org; 5Department of Nutrition, Dietetics and Food Science, Brigham Young University, Provo, UT 84602, USA; james_lecheminant@byu.edu

**Keywords:** fast mimicking, ketosis, metabolic health, glucose control

## Abstract

This study aimed to determine the impact of various fast-interrupting shakes on markers of glycemic control including glucose, β-hydroxybutyrate (BHB), insulin, glucagon, GLP-1, and GIP. Twenty-seven sedentary adults (twelve female, fifteen male) with overweight or obesity completed this study. One condition consisted of a 38-h water-only fast, and the other two conditions repeated this, but the fasts were interrupted at 24 h by either a high carbohydrate/low fat (HC/LF) shake or an isovolumetric and isocaloric low carbohydrate/high fat (LC/HF) shake. The water-only fast resulted in 135.3% more BHB compared to the HC/LF condition (*p* < 0.01) and 69.6% more compared to the LC/HF condition (*p* < 0.01). The LC/HF condition exhibited a 38.8% higher BHB level than the HC/LF condition (*p* < 0.01). The area under the curve for glucose was 14.2% higher in the HC/LF condition than in the water condition (*p* < 0.01) and 6.9% higher compared to the LC/HF condition (*p* < 0.01), with the LC/HF condition yielding 7.8% more glucose than the water condition (*p* < 0.01). At the 25-h mark, insulin and glucose-dependent insulinotropic polypeptide (GIP) were significantly elevated in the HC/LF condition compared to the LC/HF condition (*p* < 0.01 and *p* = 0.02, respectively) and compared to the water condition (*p* < 0.01). Furthermore, insulin, GLP-1, and GIP were increased in the LC/HF condition compared to the water condition at 25 h (*p* < 0.01, *p* = 0.015, and *p* < 0.01, respectively). By the 38-h time point, no differences were observed among the conditions for any of the analyzed hormones. While a LC/HF shake does not mimic a fast completely, it does preserve some of the metabolic changes including elevated BHB and glucagon, and decreased glucose and insulin compared to a HC/LF shake, implying a potential for improved metabolic health.

## 1. Introduction

Alzheimer’s disease, cancer, cardiovascular disease, and diabetes all rank among the leading causes of death in the United States [1]. These and other chronic diseases account for nearly 75% of the yearly healthcare costs in the United States, leaving many to seek behavioral and pharmacological strategies to combat them [2]. While it has long been thought that many chronic diseases were unavoidable due to genetic predispositions, we now understand that behavioral, environmental, and nutritional signals affect gene transcription and play a strong role in determining risk [3,4]. 

The implementation of dietary approaches, such as low carbohydrate diets [5] and fasting [6] protocols, may enhance metabolic health and consequent genetic expression. Wang et al. recently found that maintaining a low insulin- and low inflammation-spiking diet plays a key role in reducing life-long risk for many chronic diseases including cardiovascular disease, type 2 diabetes, and cancer [7]. Alzheimer’s disease has traditionally been deemed a genetic disease but is now popularly expressed as type 3 diabetes because of shared pathophysiological mechanisms with type 2 diabetes, namely neuroinflammation, oxidative stress, advanced glycosylation end products, mitochondrial dysfunction, and insulin resistance [8]. Even the expression of BRCA1 mutations (associated with an increased risk of breast cancer) is reduced with the consumption of a low insulinemic diet [9]. Thus, maintaining good metabolic health through healthy dietary patterns takes on a major role in mitigating the epigenetic risks of chronic diseases. 

As the link between metabolic health and chronic diseases have come into focus, it has highlighted the need to establish methods for improving metabolic flexibility. Anton et al. state that maintaining the ability to frequently and efficiently switch metabolic fuels from glucose to fatty acid-derived ketones is an indication of good metabolic health [10]. Given the typical Western dietary pattern of three or more meals per day, many individuals rarely switch their metabolic fuels, causing insulin to remain chronically elevated and ketones to be constantly low [10]. Regular fasting allows the body to make the metabolic switch to ketones more often which is commonly identified through the measurement of the ketone beta-hydroxybutyrate (BHB) in capillary whole blood [11]. 

In addition to evaluating beta-hydroxybutyrate (BHB) levels as a marker of metabolic switching and health, monitoring systemic glucose concentrations can provide insight into an individual’s metabolic and endocrine well-being [12]. Glucose (or glycemic) control refers to maintaining blood glucose levels within a specific target range. Research has shown that postprandial glucose response is an important predictor of insulin resistance and other metabolic disorders such as type 2 diabetes and cardiovascular disease [13]. When insulin resistance is present, the postprandial glucose response is typically higher and more prolonged than in those with healthy insulin sensitivity [14]. Tight glycemic control has been associated with improved overall health outcomes including reduced mortality, improved quality of life, and improved functional status [15]. To better quantify glycemic control, an increasing number of researchers are using continuous glucose monitors (CGMs) to measure the glucose response to various stimuli in both laboratory and free-living conditions [16]. These data can help identify underlying mechanisms of metabolic diseases and identify patterns and trends that indicate poor glycemic control. 

Measuring both BHB and glucose levels can provide information about the body’s response to various foods, fasting regimens, or other metabolic stressors, and valuable insights for measuring a metabolic switch. A more comprehensive understanding of glycemic control in various conditions can be recognized by measuring the hormones that regulate it, namely: insulin, glucagon, glucagon-like peptide 1 (GLP-1), and glucose-dependent insulinotropic polypeptide (GIP). One strategy used among those who wish to maintain fasting glucose and BHB levels is a fast-mimicking diet. Brandhorst et al. describes the fast-mimicking diet as a moderate to low caloric diet (consumption of less than 1000 kcal per day) consisting of moderate (34–47%) carbohydrate, low (10%) protein, and high (44–56%) fat consumption [17]. While fast-mimicking diets have been explored for their effects on weight management, blood pressure, and cholesterol, there are paucities in the scientific literature that need to be addressed which include a lack of long-term studies [18], limited research in humans [19], lack of standardized protocols [20], and unclear mechanisms of action relating to how this diet impacts glycemic control [21]. 

Work has been completed to describe the effects of a fast-mimicking diet to improve triglycerides, low-density lipoprotein cholesterol, c-reactive protein, and ketones [18,19]. Additionally, it has been found to alter genetic expression to improve pancreatic beta cell function [22], markers of aging and cancer [18], and insulin-like growth factor 1 (IGF-1). While these measurements have the potential to improve insulin sensitivity, glycemic control, and metabolic switching, they have not addressed these outcomes directly [23]. The primary purpose of this study was to compare insulin, glucagon, GLP-1, GIP, BHB, and glucose concentrations using the conditions of a water-only fast, a fast interrupted by a low carbohydrate/high fat/moderate protein shake (LC/HF), and another fast interrupted by a high carbohydrate/low fat/moderate protein shake (HC/LF) to determine the extent to which the consumption of the shakes reflects the metabolic state of fasting. Describing the specific endocrine patterns of fasting and fast mimicking in relation to glycemic control can provide tools for those seeking to improve their metabolic health. 

## 2. Materials and Methods

A randomized crossover design with counterbalanced treatment conditions was used for this study. One condition provided the participant with a LC/HF shake to interrupt the 38-h fast at 24 h of fasting. Another intervention provided the participant with a HC/LF, yet isocaloric and isovolumetric, shake to interrupt the 38-h fast at 24 h of fasting. The third condition acted as a control condition and provided the participant with an isovolumetric amount of water at 24 h of fasting. The effects of these conditions on markers of metabolism, and glycemic control were assessed. Each of the three fasts began at 6:00 p.m. and ended at 8:00 a.m., a day and a half later. Multiple studies have demonstrated water fasting up to and exceeding 38 h to be safe and well tolerated for healthy participants [24,25,26]. Approval from Brigham Young University’s Institutional Review Board was obtained prior to initiating any aspect of this study. 

Participants completed all three fasting conditions, with a 6- to 10-day washout between each session. Using randomizer.org, condition order was randomly assigned to participant numbers prior to this study [27]. A participant number was assigned to participants chronologically from the time they joined this study by signing the consent form. Prior to each laboratory session, participants were screened for contraindications to participation as outlined below. The following outcome variables were measured: body mass index (BMI), percent body fat, fat mass, capillary BHB levels, continuous interstitial glucose through a continuous glucose monitor (CGM), and plasma insulin, glucagon, GLP-1, and GIP concentrations.

### 2.1. Participants

A total of 29 healthy adults (12 female and 17 male) were recruited through word of mouth, advertisements, fliers, and social media. Table 1 reports the demographic characteristics of those participating in this study. Participants were 18 years of age or older and were weight stable (±3% body weight) for the past 3 months with a BMI between 27 and 35 [28].

Exclusion occurred if participants did not provide proper written consent or if they met any of the following exclusion criteria:Diagnosed with a chronic disease (i.e., cancer, heart/liver/kidney disease).Diagnosed with a metabolic disease (i.e., Type I and Type II diabetes).Diagnosed with an eating disorder (i.e., anorexia, bulimia or binge eating disorder).Taking medications that alter metabolism, appetite, or neurological function (i.e., insulin, metformin, amphetamine-based ADHD medications, depression, and anxiety medications such as selective serotonin reuptake inhibitors, serotonin and norepinephrine inhibitors, and benzodiazepines) [29].Food allergies (i.e.,—nuts, celiac disease, or gluten intolerance, or lactose intolerance).Habitually consumption of 60 mg or more of caffeine daily [30].Pregnant or lactating.Post-menopausal [31].Currently participating in ketogenic, carbohydrate, or calorie-restricted diets.Regularly exercised more than 225 min per week.Fasting more than once per week.Irregular sleeping patterns (including graveyard or swing shifts).

### 2.2. Measurements 

Body weight and height were measured for all participants at the beginning of each session. Weight was measured using a digital scale (Seca, Hamburg, Germany) accurate to ±0.1 kg with participants dressed in athletic shorts and a t-shirt. Height was measured by a stadiometer accurate to ±0.1 cm (Seca, Hamburg, Germany). Body mass index (BMI) was calculated as weight (in kilograms) divided by the square of height (in meters). A GE iDXA (GE, Fairfield, CT, USA) was used to assess fat-free mass, fat mass, lean mass, percent body fat, and visceral adipose tissue [32,33,34]. Visceral fat was calculated using the CoreScan application of the GE iDXA [35,36]. Calibration of the DXA scan took place at the beginning of each testing day using a manufacturer-provided calibration block. Scans were analyzed using Encore software version 17.

#### 2.2.1. Venipuncture

Blood was drawn in the Human Performance Lab by trained phlebotomists who used sterile techniques and standard phlebotomy procedures to minimize risks to the participants. One 4 mL vacuum-sealed tube prepared with ethylenediaminetetraacetic acid (EDTA) was taken from each participant at or near the median cubital vein. For processing, each 4 mL tube was inverted to allow for mixture with the EDTA. Each sample was centrifuged for 15 min at 1500 rotations per minute within 10 min of collection, after which the plasma was extracted, then placed in three separate cryovials. Ten microliters of 100× protease inhibitor cocktail were added to each milliliter of the plasma according to the manufacturer’s recommendations (Thermo Fisher Scientific, Inc., Waltham, MA, USA). Plasma samples were stored in a −80 °F freezer for future analysis. The Human Metabolic Hormone Magnetic Bead Panel multiplex kit (Cat. # HMHEMAG-34K) was used to measure concentrations of various metabolic hormones including insulin, glucagon, total GLP-1, and GIP. 

#### 2.2.2. Capillary Ketone Assessment

Capillary ketone measurements were assessed using the Precision Xtra portable ketone meter (Abbott Laboratories, Abington, UK). A 5 µL capillary blood sample was applied to an electrochemical strip inserted into the sensor to quantify BHB concentrations at 0, 24, 25, 28, and 38 h of fasting. The Precision Xtra portable ketone monitor was demonstrated by Byrne et al. to be accurate in measuring real-time whole blood capillary BHB compared to venous whole blood reference samples up to blood levels of 6 mmol/L [37]. The mean difference between sensor and reference values was +0.02 (−0.6 to +0.6) mmol/L with a reproducibility standard deviation of 0.13 [37].

#### 2.2.3. Continuous Glucose Monitoring

The Freestyle Libre Pro (Abbott Laboratories) continuous glucose monitor (CGM) was used to assess glucose levels before, during, and after each fast. Each CGM was inserted under the skin on the back of the non-dominant upper arm, per the manufacturer’s specifications, to ensure sensor accuracy [38]. The interstitial glucose concentration reflects the intravascular glucose concentration within minutes, making this a useful tool to measure the glycemic response to various stimuli [39]. The sensor measures the glucose concentrations in the interstitial compartment every 15 min and stores 14 days of data. These data were downloaded using a wireless scanner and uploaded to online cloud storage for later analysis. The Freestyle Libre Pro has been validated for accuracy and reliability compared to the Yellow Spring Instrument standard with an 11.4% mean absolute relative difference in readings over 14 days [40,41,42].

### 2.3. Procedures

Potential participants for this study were sent an email containing a link to an online survey. The online questionnaire was used to ensure participants met inclusion criteria. As part of the online survey, participants were asked to report any food allergies and complete a food preference questionnaire. The food preference questionnaire was used to ensure that participants would eat the standardized meals and shakes. Qualifying candidates were invited to participate in this study and were instructed to avoid caffeine consumption and other stimulants on the testing day as well as to refrain from vigorous physical activity for the 24-h period prior to testing. Adherence to the pre-test day protocols were assessed at the beginning of each session. If pre-test protocols had not been followed, the participant was rescheduled. 

#### 2.3.1. Orientation

Informed consent was given by participants prior to participation in any aspect of this study. Participants reported to the Human Performance Research Lab at Brigham Young University for each assessment. Each participant was informed of the main purpose of this study and were familiarized with the testing procedures. Training for proper portable ketone meter use took place in accordance with the manufacturer’s guidelines (Abbott Laboratories, Abington, UK), and participants were given a copy of these testing instructions. Using Qualtrics’s online survey software (Qualtrics.com, accessed on 8 April 2023), participants logged their own capillary ketone blood levels each time they were in the lab and at 28 h of fasting. Participants were reminded via automated text message to take and record these measurements, which contributed to a 98% recording rate. The same system was used to remind participants of appointments and was an effective way to ensure compliance in past studies [43]. Participants were oriented to the Qualtrics software (version 2023) and given login information during the initial orientation. Participants were asked to go about their normal activities of daily living during the testing period but to avoid exercise or strenuous activity including strength or cardiovascular training, yard work, hiking, or other moderate activity. Participants were also asked to maintain their normal sleeping patterns. 

#### 2.3.2. Standardized Meals

Participants were provided with a standardized meal prior to each fast. The energy needs for each participant were estimated using equations validated by Hall et al. and used by the National Institutes of Health [44], using height (cm), weight (kg), age (years), and sex to predict basal metabolic rates (BMR) [45]. An activity factor of 1.4 was used to estimate total daily energy requirements [46]. Meals were standardized based on macronutrient content (60% CHO, 25% fat, 15% protein). Participants were given 25% (BMR × 1.4 × 0.25) of their daily caloric requirements in the standardized meal. The same foods were provided on all test days and participants were instructed to consume all the food provided for each meal. Meal adherence was assessed in each session by direct observation by the researchers. 

#### 2.3.3. Standardized Shakes

Participants were provided with a standardized shake to interrupt two of the three fasts. Participants were given 25% (BMR × 1.4 × 0.25) of their daily caloric requirements in the standardized shake in two of the three fasts using the same BMR calculations as the standardized meal. The shakes were equal in total volume, protein, and fiber content, but the fat and carbohydrate composition were different. The HC/LF shake consisted of 70% carbohydrate, 20% protein, and 10% fat while the LC/HF shake consisted of 10% carbohydrate, 20% protein, and 70% fat. The protein for each shake was chocolate-flavored casein powder. The fats for each shake were entirely saturated fatty acids composed of an even mixture of unflavored medium chain triglyceride (MCT) powder and coconut oil powder. The carbohydrates for each shake consisted of unflavored monosaccharide dextrose (d-glucose) powder. Inulin fiber powder was added at 2 g for every 100 calories in the shake.

The LC/HF shake was designed to limit insulin secretion and support metabolic switching. Insulin strongly inhibits ketosis and promotes carbohydrate oxidation [47]. In addition, both shakes consisted of 20% protein to improve satiation and slow gastric emptying and absorption of macronutrients [48]. Protein does have an insulinotropic response but is more mild compared to glucose [49]. Shakes were weighed to match the caloric and macronutrient requirements of each participant using a commercial digital scale. Participants consumed the entire shake and consumption adherence in each session was assessed by direct observation by the researchers. 

#### 2.3.4. Treatment Sessions 

As outlined in Figure 1, participants were asked to report to the Human Performance Research Lab at 6:00 p.m. the night before the first fast. During this visit, consent was provided, a DXA scan was obtained, and the CGM was placed on the back of the non-dominant arm. At 5:00 p.m. the next day, participants reported to the lab for initial anthropometric assessments, a baseline venous blood draw and finger prick, and a standardized meal. Participants were instructed to eat normally leading up to the fast and to abstain from food after 4 h before the fast to normalize measured blood markers. Blood pressure was taken prior to phlebotomy. The standardized meal was consumed by 6:00 p.m., which initiated the fast. Based on random assignment, participants either consumed the HC/LF shake, a HC/LF shake, or water at 24 h of fasting (at 6:00 p.m. the day after initializing the fast). The participants remained in the lab for 1 h after consuming the shake or water and a 60-min postprandial blood draw and finger prick took place in the hopes of capturing the acute effects of the intervention. The participants then proceeded with their normal daily routine and logged their own capillary BHB levels at 28 h of fasting (10:00 p.m.). The participants returned to the lab at 7:30 a.m. the next day for a blood draw and finger prick. Once these assessments were taken, the fast could be broken and activities of daily living resumed. 

### 2.4. Statistical Analysis

The sample size for this study was calculated a priori based on a clinically meaningful difference of 25% in the area under the glucose concentration curves between conditions [50]. Using these values and setting alpha to 0.05, a sample of 28 participants was needed to yield a 90% power (effect size of 0.66). The participants were recruited until 28 people had enrolled in this study. Two participants withdrew from this study after completing only one fasting condition each, so an additional participant was recruited to make up for the loss of data, bringing our total number of participants to twenty-nine. 

The participant data are reported as means and standard deviations. To evaluate the difference in area under the curve for glucose, a repeated measures mixed-effects analysis of variance was used. The area under the treatment curve was analyzed using the trapezoidal rule with one observation per subject by treatment with area under the curve as the dependent variable. Area under the curve was computed in an attempt to represent a total response with a single number as a measure of intensity of the response. A mixed effects ANOVA was used to evaluate the difference between conditions in the area under the curve. 

Similarly, a repeated measures mixed-effects ANOVA was used to assess the main (condition and time) and interactive effects (condition by time) for each analyte. Significant main and interactive effects were subsequently evaluated using the least squared means procedure. Condition and time were the primary factors in the model. Condition had three levels (fasting with water only, fasting with a HC/LF shake, and fasting with a LC/HF shake) and time had five levels (0 h, 24 h, 25 h, 28 h, and 38 h). The F-values presented in the results represent overall interactive and main effects while the t-values represent post-hoc pairwise comparisons of least squared means. Controlling for BMI and sex was planned a priori and were included in all analyses. Statistical analyses of the data were performed using SAS software version 9.4 for Windows.

## 3. Results

Two hundred individuals applied to participate, and all were screened using the criteria outlined. As seen in Figure 2, twenty-nine individuals qualified and were randomly assigned to a condition order. Twenty-seven individuals completed all three conditions, and two individuals withdrew from this study after completing a single condition (one because of a scheduling conflict and another because they did not tolerate the fasting well). The standardized meals and shakes provided to the participants averaged 628.7 ± 102.6 kcal. 

### 3.1. Beta-Hydroxybutyrate 

There was a significant main effect of condition for BHB area under the curve (F = 31.3, *p* < 0.0001). The total area under the curves for each condition were 4.33 ± 2.26 mmol/L × hours for the HC/LF condition, 6.01 ± 2.61 mmol/L × hours for the LC/HF condition, and 10.19 ± 5.82 mmol/L × hours for the water condition. Follow-up analysis using the least squares means procedure demonstrated significant differences between the HC/LF condition and water (t = 7.77, *p* < 0.0001) and LC/HF conditions (t = 2.70, *p* = 0.0086), as well as between LC/HF and water conditions (t = 5.12, *p* < 0.0001). The water condition yielded 135.3% more BHB than the HC/LF condition and 69.6% more BHB than the LC/HF condition. The LC/HF condition yielded 38.8% more BHB than the HC/LF condition. 

Table 2 presents the means and standard deviations of BHB concentrations at each timepoint and in each condition. Additionally, it displays the statistical differences comparing each condition at each timepoint. As expected, there was a significant main effect of time for BHB concentrations (F (4, 302) = 65.42, *p* < 0.0001), and the BHB concentration increased over the course of the fast. There was also a significant condition by time interaction (F (8, 302) = 10.00, *p* < 0.0001) (see Figure 3). Follow-up analysis showed that there were no differences in BHB concentrations at baseline between HC/LF and LC/HF conditions (t = 0.11, *p* = 0.9097), between HC/LF and water conditions (t = 0.73, *p* = 0.4681), or between the LC/HF and water conditions (t = 0.62, *p* = 0.5370). Similarly, there were no differences in BHB concentrations at 24 h of fasting between HC/LF and LC/HF conditions (t = 0.65, *p* = 0.5138), between HC/LF and water conditions (t = 0.34, *p* = 0.7344), or between the LC/HF and water conditions (t = 0.32, *p* = 0.7516). Differences between conditions emerged at 25 as BHB concentrations decreased in the HC/LF condition compared to the LC/HF condition (t = 2.24, *p* = 0.0261), and compared to the water condition (t = 4.82, *p* < 0.0001) as well as in the LC/HF compared to the water condition (t = 2.61, *p* = 0.0095). BHB concentrations at 28 h were lower in the HC/LF condition compared to the LC/HF condition (t = 2.36, *p* = 0.191) and compared to the water condition (t = 6.58, *p* < 0.0001). A reduction in BHB was greater in the LC/HF condition compared to the water condition at 28 h (t = 4.25, *p* < 0.0001). The final (38 h) BHB concentrations between shake conditions did not differ (t = 0.82, *p* = 0.4112), but the HC/LF and LC/HF were both different from the water condition (t = 5.43, *p* < 0.0001 and t = 4.68, *p* < 0.0001, respectively). There was no three way interaction between condition, time, and sex (F (14, 302) = 0.65, *p* = 0.8260). 

### 3.2. Glucose

Figure 4 shows the concentrations of glucose over time starting at hour 22 of the fast. There was a significant main effect of condition for the area under the glucose curve (F (2, 73) = 18.95, *p* < 0.0001). The total area under the curves for each condition were 1272 ± 20.3 mg/dL × hours for the HC/LF condition, 1184 ± 19.6 mg/dL × hours for the LC/HF condition, and 1091 ± 20.8 mg/dL × hours for the water condition. Follow-up analysis showed that the area under the glucose curve was significant between the HC/LF condition and the water condition (t = 6.23, *p* < 0.0001), between the LC/HF condition and the water condition (t = 3.21, *p* = 0.0020), and between the HC/LF and LC/HF condition (t = 3.14, *p* = 0.0024). The differences in the area under the glucose curve was 14.2% higher in the HC/LF condition compared to the water condition, 7.8% higher in the LC/HF condition compared to the water condition, and 6.9% higher in the HC/LF condition compared to the LC/HF condition. 

The mean glucose at the beginning of all fasts was 90.87 ± 17.57 mg/dL and was not different between HC/LF and LC/HF conditions (t = 1.52, *p* = 0.1322), HC/LF and water conditions (t = 0.88, *p* = 0.3834), or LC/HF and water conditions (t = 0.62, *p* = 0.5355). By 24 h, glucose concentrations were different between HC/LF and LC/HF conditions (t = 3.28 *p* = 0.0016), HC/LF and water conditions (t = 5.18, *p* < 0.0001), and LC/HF and water conditions (t = 2.02, *p* = 0.0500). These differences persisted at 25 h (t = 24.00, *p* < 0.0001; t = 18.46, *p* < 0.0001; and t = 12.98, *p* < 0.0001, respectively), and 26 h (t = 3.02, *p* = 0.0034; t = 5.45, *p* < 0.0001; and t = 2.53, *p* = 0.0135, respectively). By 28 h, there was no difference in glucose concentrations between conditions (F (2, 77) = 0.18, *p* = 0.8380). Glucose reached a mean peak of 140.3 ± 42.2 mg/dL in the HC/LF condition, 104.0 ± 24.38 mg/dL in the LC/HF condition, and 77.17 ± 18.24 mg/dL in the water condition. There was no condition by sex interaction (F (2, 71) = 0.13, *p* = 0.8819). 

### 3.3. Hormones

Table 3 presents the means and standard deviations for each hormone and depicts differences between conditions at various timepoints of the intervention. All the measured hormones had significant condition by time interactions (see Table 3). For insulin, follow-up analysis showed that there was no differences at baseline between HC/LF and LC/HF conditions (t = 0.19, *p* = 0.8459), HC/LF and water conditions (t = 0.43, *p* = 0.6685), or the LC/HF and water conditions (t = 0.24, *p* = 0.8134). Likewise, no differences in insulin were found at 24 h of fasting (t = 0.06, *p* = 0.9537; t = 0.13, *p* = 0.8989; and t = 0.19, *p* = 0.8519, respectively). One hour after the intervention (25 h), insulin was significantly elevated in the HC/LF condition compared to the LC/HF condition (t = 3.84, *p* = 0.0002) and compared to the water condition (t = 7.00, *p* < 0.0001). Insulin was also significantly increased in the LC/HF condition compared to the water condition one hour after the intervention (t = 3.19, *p* = 0.0016). By the next morning (38 h), insulin had returned to baseline levels and no differences were noted between HC/LF and LC/HF conditions (t = 0.08, *p* = 0.9371), HC/LF and water conditions (t = 0.20, *p* = 0.8395), or the LC/HF and water conditions (t = 0.29, *p* = 0.7755) (Figure 5a). 

Like insulin, the least squares means analysis of glucagon showed no difference at baseline between HC/LF and LC/HF conditions (t = 1.00, *p* = 0.3177), HC/LF and water conditions (t = 1.08, *p* = 0.2794), or the LC/HF and water conditions (t = 0.08, *p* = 0.0333). Likewise, no differences in glucagon were found at 24 h of fasting (t = 0.66, *p* = 0.5067; t = 0.65, *p* = 0.5147; and t = 1.33, *p* = 0.1849, respectively). One hour after the intervention, (time 25 h) glucagon concentrations decreased in the HC/LF group compared to the LC/HF condition (t = 3.37, *p* = 0.0009) but not compared to the water condition (t = 1.92, *p* = 0.0554), and glucagon concentrations between the water and LC/HF condition did not differ (t = 1.45, *p* = 0.1474). By 38 h, there was no difference between HC/LF and LC/HF conditions (t = 1.06, *p* = 0.2887), HC/LF and water conditions (t = 1.48, *p* = 0.1398), or the LC/HF and water conditions (t = 0.42, *p* = 0.6720) (Figure 5b). 

Analysis of GLP-1 also demonstrated no difference between conditions at baseline [LC/HF and water conditions (t = 0.03, *p* = 0.9794), HC/LF and water conditions (t = 1.44, *p* = 0.1515), and HC/LF and LC/HF conditions (t = 1.41, *p* = 0.1589)] or after 24 h of fasting [LC/HF and water conditions (t = 0.49, *p* = 0.6271), HC/LF and water conditions (t = 0.19, *p* = 0.8497), and HC/LF and LC/HF conditions (t = 0.29, *p* = 0.7703)]. One hour after the intervention (at 25 h of fasting), GLP-1 concentrations did not differ between the HC/LF and LC/HF conditions (t = 1.42, *p* = 0.1578) or between the HC/LF and water conditions (t = 1.04, *p* = 0.2989). However, concentrations of GLP-1 were higher after 25 h in LC/HF compared to the water-only condition (t = 2.48, *p* = 0.0138). By 38 h, there was no difference between conditions (LC/HF and water conditions (t = 1.27, *p* = 0.2057), HC/LF and water conditions (t = 1.90, *p* = 0.0582), and HC/LF and LC/HF conditions (t = 0.65, *p* = 0.5152)) (Figure 5c). 

Analysis of GIP revealed no difference between conditions at baseline between the LC/HF and water conditions (t = 1.58, *p* = 0.1165) and no difference between HC/LF and water conditions (t = 1.63, *p* = 0.1038), but did reveal a difference at baseline between the HC/LF condition and the LC/HF condition (t = 3.19, *p* = 0.0016). By 24 h of fasting, there were no differences in the concentrations of GIP between the LC/HF and water conditions (t = 0.14, *p* = 0.8895), no difference between the HC/LF and water conditions (t = 0.23, *p* = 0.8144), and no difference between the HC/LF condition and the LC/HF condition (t = 0.10, *p* = 0.9227). One hour after the intervention (at 25 h of fasting), GIP was higher in the HC/LF condition compared to both the LC/HF (t = 2.27, *p* = 0.0244) and the water condition (t = 9.96, *p* < 0.0001). Likewise, GIP concentrations were elevated in the LC/HF condition compared to the water condition at 25 h (t = 12.13, *p* < 0.0001). By 38 h, there was no difference between any of the conditions (LC/HF and water conditions (t = 0.33, *p* = 0.7389), HC/LF and water conditions (t = 0.36, *p* = 0.7188), and HC/LF and LC/HF condition (t = 0.03, *p* = 0.9737)) (Figure 5d).

### 3.4. Perceived Difficulty of the Fast

At the end of each fast, we asked the participants to rank the difficulty of that condition on a scale of 1 to 10 (1 being extremely easy and 10 being extremely difficult). The LC/HF was ranked as a 5.1 ± 2.09, the water-only fast as a 5.5 ± 1.75, and the HC/LF ranked as a 6.2 ± 1.33. However, these means were not statistically different (F (2, 73) = 1.52, *p* = 0.2088). Beyond verbal confirmation, adherence to fasting protocols was evaluated by observation of the CGM data and no large glucose spikes were observed during the fasting period of any individual. 

## 4. Discussion

The results of this study showed that the greatest overall decrease in BHB concentration after the fast-interrupting shakes was observed during the HC/LF condition with a smaller decrease during the LC/HF condition. Additionally, BHB concentrations in both fed conditions were lower compared to persistent fasting. The LC/HF condition continued to have elevated BHB even after consumption of the shake, but levels were slightly suppressed and returned to greater than 0.5 mmol/L by hour 38 (14 h post intervention). The HC/LF condition, on the other hand, saw the BHB concentration drop to levels near baseline after the shake consumption. However, significant BHB was again accumulating in the blood by hour 38, although this group did not recover BHB above 0.5 mmol/L by the end of the fast. Because metabolic switching is important for regulating cellular metabolism and preventing age-related diseases [51], maintaining a portion of BHB through a LC/HF shake is beneficial for maintaining the metabolic switch when interrupting a fast.

High glucose variability (large and frequent fluctuations in glucose levels) has been associated with negative health outcomes such as increased risks of retinopathy [52], cardiovascular disease [53], metabolic disorders [54], and all-cause mortality [55]. Fasting is one strategy to reduce glucose variability, and the present study builds on this by demonstrating a lower glucose variability when breaking a fast with a LC/HF shake compared to a HC/LF shake. The results for glucose followed a similar pattern as observed with BHB. LC/HF outperformed HC/LF but did not completely mimic a fasting condition [56]. Glucose increased in both conditions after consuming the shakes and remained elevated for 4 h, after which levels returned to concentrations similar to the water-only fast.

These results differ from a study conducted by Huang et al. who evaluated the impact of breaking a fast with fast-mimicking food on glucose and BHB concentrations [57]. This recently published study used a low protein, low carbohydrate, and high fat nutrition bar to break an overnight 15 h fast and monitored capillary glucose and BHB hourly for 4 h. They found no difference between water fasting and the consumption of the nutrition bar on glucose and BHB concentrations.

While the nutrition bars used in the study (9% protein, 77% fat, and 14% carbohydrate) had a similar percent of fat and carbohydrate compared to the LC/HF shake administered in this study, the bars provided fewer calories (200 kcal) and a lower percentage of calories from protein. Additionally, this study administered the nutrition bar in the morning after 15 h of fasting; in contrast, our study broke the fast at dinnertime after 24 h of fasting. The timing of shake administration is important since several studies have indicated that people are more glucose tolerant in the morning compared to the evening [58]. In addition, at 15 h, the metabolic switch to greater ketone production is just beginning and plasma ketone levels are generally only modestly elevated.

Knowing their impact on glycemic control and metabolic switching, our lab recently reported the effects of beginning a fast with and without exercise on concentrations of insulin, GLP-1, and GIP, noting that insulin and GIP reach minimal concentrations by 12 h of fasting and remain low regardless of whether exercise was used to initiate the fast [59]. We also reported that GLP-1 was elevated for up to 36 h after exercise compared to the non-exercise group. To further explore the relationship between fasting and these hormones, the current study measured insulin, glucagon, GLP-1, and GIP because of their regulatory effects on substrate utilization and influence on metabolic switching and glucose homeostasis [60].

Insulin secretion is regulated by several factors, including blood glucose levels, amino acids, and incretin hormones such as GLP-1 and GIP [61]. A recent review by Fanti et al. highlights that insulin secretion is suppressed in both fasting and fast-mimicking diets to help maintain blood glucose levels [17]. The results of our study support this evidence and demonstrate that insulin remains very low during a fasted state and increases only slightly with a LC/HF shake. Because maintaining low insulin and reducing insulin spikes have been shown to reduce the incidence of insulin resistance, and its many deleterious downstream health effects [62], a LC/HF shake could be a useful means of breaking a fast and extending some of the metabolic benefits.

Like insulin, glucagon plays an important role in glycemic control. Glucagon secretion is stimulated by low blood glucose and suppressed by high blood glucose, amino acids, and somatostatin [60]. Our lab recently demonstrated that glucagon rises steadily over the course of a 36-h fast [43], and while we suspected glucagon to react similarly in a fast-mimicking diet, this has not been described in the literature. The results of this study demonstrate that the LC/HF shake did not alter the concentrations of glucagon and there was no difference between the water-only and the LC/HF conditions at any timepoint, making the LC/HF shake fast-mimicking in this regard. The HC/LF shake, on the other hand, suppressed the concentrations of glucagon. 

GLP-1 is an incretin hormone secreted by the L cells of the small intestine in response to nutrient ingestion and plays a crucial role in glycemic control [63]. GLP-1 promotes insulin secretion, inhibits glucagon secretion, slows gastric emptying, and signals satiety to reduce appetite [64]. GLP-1 and GIP have a combined ability to regulate approximately 60% of insulin release after a meal [65]. During fasted conditions, endogenous GLP-1 concentrations are typically low [66] but rise as fasting persists [43]. However, we could not find any studies that have evaluated the response of GLP-1 to a fast-mimicking meal. The current study demonstrated that a LC/HF shake increases concentrations of GLP-1. There was no difference in GLP-1 between the HC/LF shake and water-only fast. These results are supported by Rizi et al. [67] and Gibbons et al. [68], who describe an attenuated increase in GLP-1 in response to the presence of high amounts of carbohydrates in meals.

GIP is another incretin hormone secreted by the K cells of the small intestine in response to nutrient ingestion. Like GLP-1, it promotes insulin secretion, but also promotes glucagon secretion, which can counteract its insulinotropic effects [69]. In a study of healthy individuals, the co-infusion of GIP with glucose was found to enhance insulin secretion, but also increased glucagon secretion and hepatic glucose output, resulting in a blunted glucose-lowering effect, suggesting that GIP has a more nuanced role in glycemic control than GLP-1 [70].

Studies have demonstrated that GIP decreases over the course of a fast and responds strongly to meal ingestion [71]. While GIP concentrations have not been measured in human studies in a fast-mimicking state, Yoder et al. found that GIP concentrations increase in a dose-dependent manner to lipids and carbohydrates, but not protein when directly administered via feeding tubes in rats [72]. Our study agrees with past work by affirming that GIP decreases over the course of a fast and adds to the literature in describing how it responds to a LC/HF shake. The presence of carbohydrate and fat in the shakes likely increased GIP concentrations above fasting baseline levels for a short time. While it is well established that the GIP excursion is greater in response to a high fat meal compared to a high carbohydrate meal [73], it is important to note that this long-chain fatty acids and monounsaturated fatty acids are the strongest drivers of this secretion; medium-chain fatty acids and saturated fatty acids have minimal impact on GIP secretion [74,75]. Because the composition of fat in this study was entirely medium-chain saturated fatty acids, the LC/HF shake did not elicit a larger GIP excursion than the HC/LF shake. Thus, in terms of GIP concentrations, a LC/HF shake does not mimic a fasted state. 

Despite differences in these various biomarkers between conditions, the tolerability of each fast was not different between conditions. While individual responses varied, the water-only fast was not reported to be any more difficult than interrupting the fast. Additionally, while the aim of this study was to determine the extent to which a LC/HF or HC/LF shake reflected the metabolic state of fasting and acute changes were noted across many of the variables, it is important to note that recovery was achieved by 14 h of additional fasting as concentrations of insulin, glucagon, GLP-1, and GIP were not different from each other by 38 h. Thus, while a HC/LF shake had a more profound interruption to the mechanisms regulating glycemic control than the water or LC/HF shake, the effects were short-lived. 

### Limitations and Strengths

Several limitations should be considered when interpreting the results of this study. First, although we measured the variables described at 0, 24, 25, and 38 h of fasting, the data between each of these points was not measured and the results may not be linear. Repeating a similar study with more frequent data collection would improve the understanding of how these hormones fluctuate over time. Second, each participant abstained from food for 4 h before presenting to the laboratory, but we did not control the food intake earlier that day. The amount and type of food and drink consumed before the fast and standardized meal may have affected the metabolic state. However, we asked participants to follow normal dietary patterns and to not overconsume in preparation for the fast. Additionally, baseline glucose, BHB, glucagon, and insulin did not differ between conditions, suggesting that the participants entered the conditions in a similar metabolic state each time. Third, hydration was encouraged but not directly monitored. Work by Johnson and Passmore found that hydration status plays a role in ketone production and that dehydration may hinder ketone production [76]. Finally, these results are limited to the shakes used in this study. Compared to solid food, liquid shakes are digested easier and absorbed quicker, which has specific implications for glycemic control. However, using solid foods is more complicated as the interaction between nutrients impacts digestion and alters glycemic control. Additionally, MCT and coconut oil powders are primarily made up of saturated fats and different results may have been observed by using other oils with greater concentrations of additional fatty acid types. Medium-chain triglycerides also convert more readily to ketones compared to long-chain fatty acids, which may have increased BHB concentrations. Likewise, we used casein as the protein source for the shakes. This powder was selected with the expectation of prolonged satiety, as it is digested at approximately half the rate of whey protein [77]. Using this protein source may have dampened the glucose (and subsequent insulin) spikes observed after shake consumption. It is also important to note that the carbohydrate source for this study was dextrose, which can be absorbed and used quickly in the body compared to sucrose or fructose because of the need for fructose to be metabolized in the liver. Finally, we recognize that the additional measurement of the growth hormone–insulin-like growth factor-I axis would provide stronger insights to nutrient utilization [78], and that the additional measurement of pro-inflammatory cytokines would strengthen indications of postprandial cardiovascular risk [79]. While the inclusion of these measurements would be beneficial, their absence in these outcomes does not diminish the overarching message of this study regarding glycemic control. 

This study encompasses several distinct strengths that enhance the understanding of glycemic regulation under fasting conditions. Specifically, the reactions of glucagon and GLP-1 to a LC/HF shake are described and put into the broader framework of previously described biomarkers GIP, insulin, glucose, and BHB. This study also utilized continuous glucose monitors to describe the fluctuations of glucose with more clarity in these conditions since measurements were taken every 15 min over the course of the intervention. The crossover design of this study also strengthens the ability to make within-group comparisons and identify differences between conditions while mitigating confounding variables. The study design also used a 38-h fast control condition and two additional fasts that allowed for 14 h of additional fasting after the intervention to enhance understanding of glucose recovery in these conditions. Furthermore, this study offers fresh insights to what has previously been anecdotal evidence, affirming that physiological signals of satiety may increase by consuming a LC/HF shake, potentially enhancing the sustainability of fasting while mitigating the metabolic effects of breaking a fast.

## 5. Conclusions

Individuals seeking to improve their glycemic control through fasting may consume a LC/HF shake and relieve some degree of physiologic hunger with less impact on glycemic patterns compared to a HC/LF shake. The LC/HF shake is not completely fast mimicking as it disrupts ketone production and results in a modest elevation in blood glucose. At the same time, the LC/HF shake preserves most of the metabolic changes from fasting better than a HC/LF shake. Using a LC/HF shake to interrupt a fast may be used to reduce hunger during fasting while acutely maintaining tight control of glucose, BHB, insulin, and glucagon.

The results from this study can benefit individuals who participate in intermittent fasting or time-restricted eating and want to optimize their fast for metabolic health. We recognize that 10% of the calories in the HC/LF shake came from glucose, and additional work could be completed to determine if BHB, glucose, and the hormones measured would approach a more fasted-like state if glucose was decreased further. Future work should also seek to understand the long-term sustainability of these dietary patterns and their effects on glycemic control over time, and additional attention should be given to uncovering the mechanisms that accompany these approaches to fasting and their combination with exercise and/or pharmaceuticals for optimizing glycemic control. 

## Figures and Tables

**Figure 1 nutrients-16-00164-f001:**
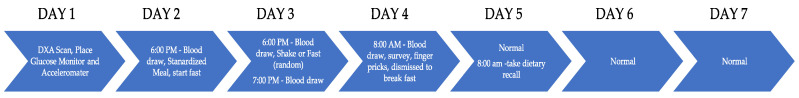
Weekly outline of intervention.

**Figure 2 nutrients-16-00164-f002:**
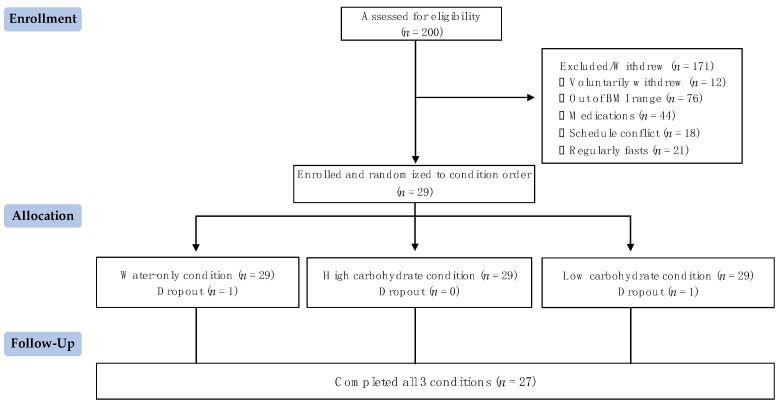
Participant flow diagram.

**Figure 3 nutrients-16-00164-f003:**
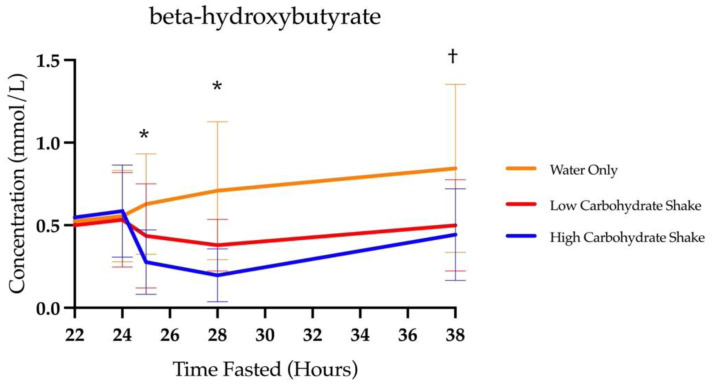
Concentrations of BHB over time with focus on time of intervention and beyond. * All conditions are different from each other. † Low carbohydrate and high carbohydrate differ from water, but not each other.

**Figure 4 nutrients-16-00164-f004:**
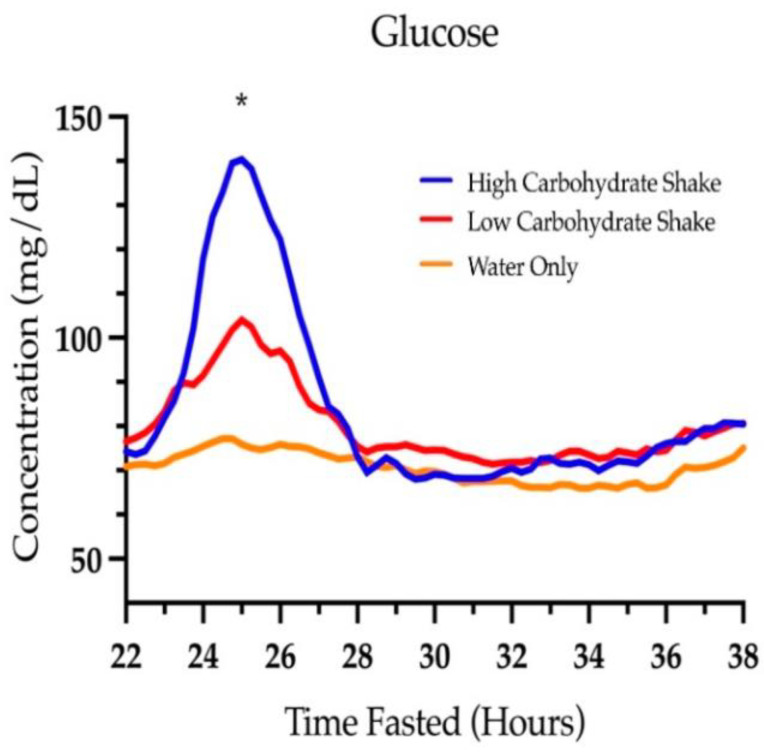
Glucose concentrations over time. * Signifies a difference between all three conditions (*p* < 0.05).

**Figure 5 nutrients-16-00164-f005:**
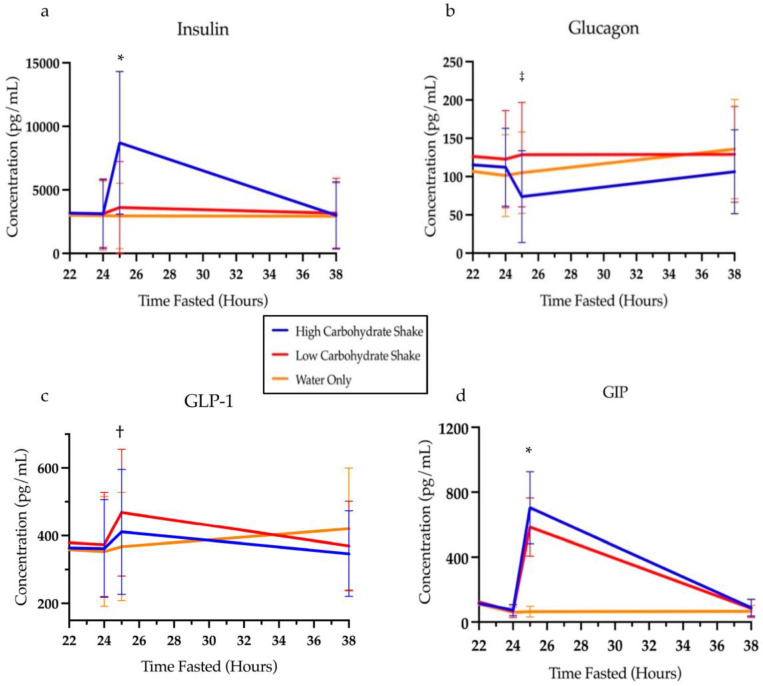
Concentrations of insulin (**a**), glucagon (**b**), GLP-1 (**c**), and GIP (**d**) over time. * Signifies differences between all conditions at these time points (*p* < 0.05). ^†^ Signifies differences between LC/HF and water condition at this point (*p* < 0.05). ^‡^ Signifies differences between HC/LF and LC/HF conditions at this point (*p* < 0.05). HC/LF = high carbohydrate/low fat; LC/HF = low carbohydrate/high fat.

**Table 1 nutrients-16-00164-t001:** Demographic characteristics of participants.

	Male (*n* = 17)		Female (*n* = 12)		Cumulative (*n* = 29)	
	Mean	SD	Mean	SD	Mean	SD
Age (years)	36.2	15.9	35.8	8.9	36.0	13.3
BMI (kg/m^2^)	31.8	4.6	30.5	4.5	31.2	4.5
Percent body fat	30.5	8.14	41.5	5.2	35.4	8.8
Visceral Adipose (g)	1238.8	916.1	953.6	502	1120.8	773.9
Ethnicity	*n*	%	*n*	%	*n*	%
African	2	11.8	1	8.3	3	10.3
Caucasian	13	76.4	8	66.7	21	72.4
Hispanic/Latino	2	11.8	3	25	5	17.3

**Table 2 nutrients-16-00164-t002:** BHB concentrations (mmol/L) over time and between conditions.

	0 h *		24 h *		25 h ^†^		28 h ^†^		38 h ^‡^	
Condition	Mean	SD	Mean	SD	Mean	SD	Mean	SD	Mean	SD
HC/LF	0.13 ^a^	0.05	0.59 ^b^	0.28	0.28 ^c^	0.19	0.19 ^a,c^	0.16	0.44 ^d^	0.28
LC/HF	0.14 ^a^	0.07	0.53 ^b^	0.29	0.44 ^b,c^	0.16	0.38 ^c^	0.16	0.51 ^b^	0.27
Water	0.18 ^a^	0.20	0.56 ^b^	0.28	0.63 ^b,c^	0.31	0.70 ^c^	0.42	0.85 ^d^	0.051

A significant condition by time interaction was present for all conditions (F = 10.09, *p* < 0.0001). ^a–d^ Indicates a significant difference between time points in the given condition (*p* < 0.05). Means with the same letter on the same row were not significantly different. * No difference between means for all three conditions in the same column (time point). ^†^ All three means in the same column (time point) are significantly different (*p* < 0.05). ^‡^ HC/LF and LC/HF are significantly different than water only (*p*’s < 0.05) but not different than each other. HC/LF = high carbohydrate/low fat; LC/HF = low carbohydrate/high fat.

**Table 3 nutrients-16-00164-t003:** Hormone concentrations (pg/mL) at various time points in each condition.

**Analyte**	**Condition**	**0 h**	**24 h**	**25 h ***	**38 h**	**F-Value**	***p*-Value**
Insulin	Water	3414.4 ± 2635.7 ^a^	2969.7 ± 2730.5 ^a^	2955.4 ± 2578.5 ^a^	2932.9± 2610.7 ^a^	18.6	<0.0001
	LC/HF	3614.4 ± 2618.7 ^a^	3129.8 ± 2650.2 ^b^	3612.8 ± 3612.8 ^a^	3172.6 ± 2749.1 ^b^		
	HC/LF	3803.1 ± 3297.8 ^a^	3112.7 ± 2744.0 ^a^	8706.7 ± 5615.2 ^b^	3011.8 ± 2631.1 ^a^		
**Analyte**	**Condition**	**0 h** **^‡^**	**24 h**	**25 h ***	**38 h**	**F-Value**	***p*-Value**
GIP	Water	660.7 ± 420.2 ^a^	60.2 ± 32.7 ^b^	64.1 ± 33.3 ^b^	66.8 ± 35.8 ^b^	22.1	<0.0001
	LC/HF	743.4 ± 317.2 ^a^	67.8 ± 68.9 ^b^	585.1 ± 178.9 ^c^	84.6 ± 56.0 ^bd^		
	HC/LF	575.1 ± 314.2 ^a^	73.2 ± 34.7 ^b^	704.9 ± 222.9 ^c^	88.0 ± 51.4 ^d^		
**Analyte**	**Condition**	**0 h**	**24 h**	**25 h** **^ƒ^**	**38 h**	**F-Value**	***p*-Value**
GLP-1	Water	442.9 ± 170.1 ^a^	353.3 ± 161.7 ^b^	367.7 ± 159.5 ^bc^	420.4 ± 179.7 ^c^	4.1	0.0006
	LC/HF	442.3 ± 144.8 ^a^	373.4 ± 153.8 ^b^	468.3 ± 187.6 ^ac^	369.6 ± 132.3 ^cd^		
	HC/LF	385.4 ± 122.5 ^a^	362.2 ± 144.4 ^b^	411.2 ± 184.7 ^b^	347.0 ± 126.6 ^b^		
**Analyte**	**Condition**	**0 h**	**24 h**	**25 h** **^†^**	**38 h**	**F-Value**	***p*-Value**
Glucagon	Water	167.1 ± 66.3 ^a^	101.3 ± 53.4 ^b^	105.2 ± 53.3 ^b^	135.9 ± 64.9 ^c^	3.4	0.0029
	LC/HF	165.8 ± 58.9 ^a^	122.9 ± 63.5 ^b^	128.8 ± 68.3 ^b^	128.9 ± 62.6 ^b^		
	HC/LF	149.6 ± 61.5 ^a^	112.2 ± 50.8 ^b^	73.89 ± 60.1 ^c^	106.37 ± 54.7 ^b^		

Mean ± standard deviation. F and *p*-values referred to the condition by time interaction for each outcome. * All three means in the same column (time point) were significantly different (*p* < 0.05). ^†^ LC/HF differed from HC/LF (*p* < 0.05), but neither differed from water. ^‡^ Water and HC/LF were the same, but different from LC/HF (*p* < 0.05). ^ƒ^ Water and LC/HF were different (*p* < 0.05), but not different between the other condition. ^a–d^ Indicates a significant difference between time points in the given condition (*p* < 0.05). Means with the same letter on the same row were not significantly different. HC/LF = high carbohydrate/low fat and LC/HF = low carbohydrate/high fat.

## Data Availability

The data presented in this study are openly available in OpenScience Framework at doi: 10.17605/OSF.IO/9HJ8G.

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
