# Peer review of "The Effects of a High-Carbohydrate versus a High-Fat Shake on Biomarkers of Metabolism and Glycemic Control When Used to Interrupt a 38-h Fast: A Randomized Crossover Study"

_nutrients, 2024, doi:10.3390/nu16010164_

Round 1

Reviewer 1 Report

Comments and Suggestions for Authors

In this manuscript, Deru et al. examined the effects of fasting and interruption of fasting on the levels of beta-hydroxybutyrate (BHB), glucose, insulin, glucagon, and incretin hormones. Although the observed changes in BHB, glucose, and hormone levels are largely as expected, their cross-over design of fasting with/without interruption in 27 sedentary adults provided important reference data for future physiological studies focusing on fasting/refeeding. To strengthen their argument, especially on fuel switching, I would raise the following points:

1. In lines 239-257, can you provide a more detailed composition of the shakes? Particularly, the composition of fat (saturated/unsaturated fatty acid compositions and cholesterol) should influence incretin hormone secretion. Also, the composition of carbohydrates (glucose, other monosaccharides, disaccharides, etc) should also be provided.

2. GLP-1 measurement. In lines 175-178, did you measure active or total GLP-1? Also, plasma GLP-1 concentrations shown in Table 3 and Fig. 5 are quite high. Could you make sure that these values/units are correct? 

3. GIP values in Table 3 and Fig. 5d are intriguing because GIP is generally more potently induced by a fat-rich meal than by a simple glucose load. Can you provide some argument on the reason why GIP is similarly strongly induced by HC/LF and LC/HF shakes?

4. To argue metabolic fuel switching, the growth hormone (GH)-IGFI axis is also critical. GH plays an essential role for the protein-sparing effect during fasting, enhancing lipolysis and gluconeogenesis. Can you measure GH using the remaining plasma samples?

5. Several studies suggested that postprandial glucose spike is a cardiovascular risk and glucose spike may be pro-inflammatory. Can you compare some representative proinflammatory cytokines (e.g., IL-1beta) between HC/LF and LC/HF postprandial (post-fasting interruption) samples?

Author Response

Please see the attached document for details of revisions.  

Reviewer 2 Report

Comments and Suggestions for Authors

With the increasing attention to improve metabolic health for combating chronic diseases, this paper attempted to address the question about the impact of various fast-interrupting shakes on markers of glycemic control. It is a very interesting topic, however there are some concerns over the presentation of the experimental procedure and results. 

Major concern: The presentation of the rationale for selection of statistic analysis and the presentation of the statistical analysis result need to be improved.

Please specify what statistical analysis package/software was used?  

The F and t scores used in the paper need to be specified/defined. 

How the area under the curve was calculated? 

Author Response

Please see attached for comments and revisions.

Thank you for your time and efforts. 

Round 2

Reviewer 1 Report

Comments and Suggestions for Authors

The authors have sufficiently addressed my points and the revised manuscript is suitable for publication.

Reviewer 2 Report

Comments and Suggestions for Authors

No more concerns.